# Online Teaching and Learning in Higher Education during the Coronavirus Pandemic: Students' Perspective

**Claudiu Coman** [1] , **Laurențiu Gabriel Țîru** [2], **Luiza Meseșan-Schmitz** [1,*] , **Carmen Stanciu** [2] **and Maria Cristina Bularca** [1]

[1] Faculty of Sociology and Communication, Transilvania University of Brasov, 500036 Brașov, Romania; claudiu.coman@unitbv.ro (C.C.); maria-cristina.bularca@student.unitbv.ro (M.C.B.)

[2] Faculty of Psychology and Sociology, West University of Timisoara, 300223 Timișoara, Romania; laurentiu.tiru@e-uvt.ro (L.G.Ț.); carmen.stanciu@e-uvt.ro (C.S.)

\* Correspondence: luiza.mesesan@unitbv.ro

**Abstract:** The research focuses on identifying the way in which Romanian universities managed to provide knowledge during the Coronavirus pandemic, when, in a very short time, universities had to adapt the educational process for exclusively online teaching and learning. In this regard, we analyzed students' perception regarding online learning, their capacity to assimilate information, and the use of E-learning platforms. An online survey based on a semi-structured questionnaire was conducted. Data was collected from 762 students from two of the largest Romanian universities. The results of the research revealed that higher education institutions in Romania were not prepared for exclusively online learning. Thus, the advantages of online learning identified in other studies seem to diminish in value, while disadvantages become more prominent. The hierarchy of problems that arise in online learning changes in the context of the crisis caused by the pandemic. Technical issues are the most important, followed by teachers' lack of technical skills and their teaching style improperly adapted to the online environment. However, the last place was assigned by students to the lack of interaction with teachers or poor communication with them. Based on these findings, research implications for universities and researchers are discussed.

**Keywords:** online teaching; E-learning platform; higher education; students' preferences; online information assimilation

## 1. Introduction

The coronavirus pandemic has generated changes in the teaching-learning process in higher education institutions and has influenced the interaction between teachers and students. As a consequence of the pandemic, universities were constrained to carrying out their activity with students exclusively online [1]. In this regard, many governments took measures in order to avoid spreading the virus and to ensure the continuity of the educational process, and universities worldwide adopted online learning [2].

While in general, internet-based learning is considered an option, an alternative to traditional learning [3], during the Coronavirus pandemic it became an essential element for maintaining the activity of schools and universities. This paradigm shift could generate changes in students' perception of this way of teaching and their perception might be different from the one found in studies previous to the pandemic. Thus, through this paper, we tried to capture the existence of such changes.

Previous studies show that E-learning offers many benefits for students because this type of learning involves student-centeredness, it is more flexible [4], and it can also improve interaction

with students by providing asynchronous and synchronous tools such as e-mail, forums, chats, videoconferences [5,6]. Furthermore, internet technologies facilitate the distribution of content at the same time, to a large number of users; E-learning platforms offer many advantages to learners such as control over the content, control over the time spent learning, and thus the process can be adapted according to the learner needs and objectives of learning [7]. This might contribute to better communication with the students and in spite of some inherent challenges brought by this time of crisis, E-learning might enhance the learning process for students.

However, when using E-learning platforms there are also some elements that might be considered obstacles in students' process of learning, such as decreased motivation in students, delayed feedback or help due to the fact that teachers are not always available at the time students may need help while learning, or feelings of isolation due to lack of physical presence of classmates [8]. Nonetheless, these obstacles can be overcome with the help of teachers who should adapt their teaching strategies to the needs of students. In order to do so, experience and knowledge about teaching in the online environment are necessary. Thus, we believe that these challenges and disadvantages could be more prominent while the educational process takes place exclusively online. This might happen due to the lack of teachers' experience in using E-learning and due to the short time in which they had to adapt their teaching style to the new conditions. Relevant in this way are the results of a study conducted by School Education Gateway at the beginning of the pandemic which showed that 66.9% of respondents affirmed that they used online platforms for teaching for the first time [9]. Thus, it can be inferred that students and teachers were not ready for an entirely online experience.

Therefore, both university members and students came across many challenges. The Organization for Economic Co-operation and Development mentioned that some of the challenges universities have to face were: keeping an equilibrium between online courses, that could affect students health, them spending many hours in front of a screen, and non-digital activities, analyzing and focusing on student's emotional health—providing them with support throughout the process of learning, taking into account the fact that not all students have access to the internet, and managing and monitoring their access to devices in order to effectively collaborate with them [10]. Furthermore, universities also struggle with keeping the content of the course consistent and relevant, with communicating clearly with the academic community, and also with acquiring and recruiting students [5]. On the other hand, students also had to face challenges and a study focusing on students' perspective on E-learning identified that among the main challenges that students encountered were accessibility, connectivity, lack of appropriate devices, social issues represented by the lack of communication and interaction with teachers and peers [11].

Taking into account the aspects mentioned above we believe that the transition to exclusive E-learning can highly affect the educational process and student's perception about the use of the online environment in the process of teaching and learning, and these ideas stand at the basis of our research. We considered it important, relevant, and necessary to analyze whether students have adapted to E-learning and whether they are satisfied rather than dissatisfied with this exclusive online experience.

The aim of the paper is to identify the student's perspective on the E-learning experience during the Coronavirus pandemic in order to improve and strengthen the E-learning system. In other words, our study examines the way the process of learning was affected during this time of crisis and investigates the student's view on the use of E-learning platforms and how these technologies influenced their understanding and assimilation of information. Furthermore, we were also interested in identifying the main difficulties that students encountered while learning online, and in which context they would like these platforms to be further used by universities.

Thus, our research can contribute to the development of the E-learning process, for it provides information on the usefulness of certain methods used in order to deliver the courses, the time spent on completing tasks and projects, the content of the course, and also student's grievances, recommendations and preferences for teaching techniques.

## 2. Literature Review

### 2.1. E-Learning in Higher Education

Nowadays, the higher education system is in a continuous process of change, universities having to keep pace with the needs, desires, and requirements of students. Thus, information technologies and E-learning systems are seen as essential factors in carrying out the activity of universities, these institutions investing more and more in online systems and devices [12]. However, in the technology era, one of the main challenges of universities is the integration of innovative E-learning systems so as to reinforce and support both teaching and learning [13].

Due to its complexity, multiple definitions are proposed for the concept of E-learning. In a simple way, E-learning means using information and computer technologies and systems in order to build and design learning experiences [14]. Similarly, Elmarie Engelbrecht describes E-learning as a concept that uses electronic media represented by the internet, CD-s, mobile phones, or even television, in order to provide distance learning and teaching [15]. In short, E-learning refers to transferring knowledge and education by utilizing various electronic devices [16], and the concept can be better understood when is integrated into a context in which technology is used in order to meet people's need to learn and evolve [17].

Early forms of distance education date back to 1840, when Isaac Pitman used mail and a shorthand technique in order to teach and collaborate with students [18], and it is taught that the term E-learning began to be used in the educational field in the mid-1990s [19]. Taking the previously mentioned aspects into account, this type of online learning can be viewed as a natural development of the concept of distance learning [20]. A more complex and inclusive definition states that E-learning can be considered a particular form of teaching and learning, that integrates electronic resources and mediums whose role is to foster development and to make education and training more qualitative [21]. E-learning is also viewed as a system used for formal teaching, or a network where information is sent through electronic resources to a large audience. The main elements that ensure the functioning of such systems are computers and the internet [22].

Offering a wide range of possibilities for sharing information and uploading documents with different formats, E-learning has certain features that facilitate and nurture the learning-teaching process. Because it is a web-based system, the installation of additional tools is not required, and once uploaded, the content is available for users at any time [23]. In this regard, the variety of technological tools that are available today allowed the development of many types of E-learning. Some of these types were identified by Horton, and are represented by individual courses, that people take on their own without having classmates, virtual classes, that are constructed similarly to a traditional, face to face course, learning games, where the process of understanding and assimilating information is done through activities that are simulated, blended learning, that combines traditional and online classes, mobile learning, or knowledge management, which refers to the online distribution of documents and materials that are meant to educate not just individuals, but large numbers of people, communities, and organizations [13]. Thus, being a complex process, E-learning includes elements such as technological tools and design, e-learning platforms, content, and users/ participants [17]. E-learning differs from traditional or other methods of learning because, according to Oye et al., it does not only focuses on instruction but also on learning that is adjusted to individuals [24]. In other words, while traditional education is more teacher-centered, with the development of E-learning a shift towards a student-centered education can be seen [25].

Differences between traditional and online learning may also be acknowledged in terms of principal sources of information, assessment, or quality of education. While in traditional education, students are evaluated only by teachers, who also represent their main source of information, and the quality of education is dependent on teacher's knowledge and skills, in online learning, the evaluation may be done with the help of tools and systems, students can procure information from various documents uploaded on the platform, and the quality of education is influenced by the level of training that teachers

have in using technology, and also their teaching style [26]. Cheung and Cable identified and described eight principles that stand at the core of effective online teaching, such as: encouraging contact between students and faculty, collaborative learning, quick feedback, active learning, task time—encouraging students to allocate more time for completing tasks, high-expectations—the teacher should communicate their expectations in order to encourage and motivate students, diversified learning, and technology application [27].

Considering that, the evolution and use of systems and technologies favored the development and expansion of educational opportunities [28], the use of E-learning in higher education and student's perception of the usefulness of this type of learning became subjects of interest for many researchers.

Relevant in researching the use of E-learning is the Technology Acceptance Model (TAM), which proved to be helpful in analyzing and comprehending the way students intend to use E-learning [29]. The model was developed by Fred Davis, who believed that the extent to which people accept the integration of technology can be an essential factor for the success of information systems. The model provides information and explains the relations behind the features of a system, the way people behave while using it, and the attitude that people may have towards using the system—which is influenced by perceived usefulness and ease of use [30].

A study [31], focusing on student's perception on the implementation and integration of E-learning platforms while using TAM model as a theoretical background, revealed that all students were of the belief that the E-learning module they took was useful and easy to use, stating that they understood information, and navigated and accessed documents effortlessly. A similar study based on the TAM model and developed at the University of Jordan [29], confirmed that both perceived usefulness and ease of use directly influence the attitude that students have towards using E-learning. Furthermore, TAM was also used to investigate teachers' perception of E-learning, a study [32] showing that together with their previous experience, the perception teachers had regarding E-learning affected their behavior and the way they actually use it.

With regard to the use of E-learning in higher education, generally, the literature provides results in favor of its usefulness, effectiveness, and positive influence on student's performance. According to a study on the impact of E-learning on students and teachers [33], most of the respondents, represented by teachers, believe in the potential of E-learning to enhance the educational process and affirm that it improves collaboration and communication with students, and that it offers flexibility and helps students to better understand the lectures. Investigating students' attitude towards E-learning, Dookhan revealed that their attitude was positive and that it improves when they perceive that E-learning systems are easy to access [34]. Another study [35] pointed out that, when used as an additional method to traditional classes, E-learning enhanced students' learning experience and increased their engagement with the lectures. A research focused on comparing traditional with online learning [36] showed that a high percentage of the students who completed the survey stated that they have assimilated more information in face-to-face classes than online, but they positively perceived their overall online experience, even though they have encountered difficulties while using E-learning platforms.

However, while most studies highlight positive attitudes towards E-learning, similar studies concluded that students were of the opinion that online courses do not have the same value as courses taught in the classroom [37], and that students would rather accept blended learning, a combination of online and face to face classes, rather than only online learning [38].

## 2.2. E-Learning Platforms in Higher Education

The E-learning process in higher education is done with the help of various online platforms. Over time, many notions were used to describe online learning, such as Computer-mediated learning [39], Web-based training, E-learning systems, and Learning Management Systems [40]. Regardless of their name, all these systems have the use of the Internet in common, and certain features that allow registration, assessment of the activities of learners and teachers [40], and that also facilitate

the delivery of lectures and interaction between students, their colleagues and teachers. Among the most important functions of online learning platforms are forums that allow student-teacher communication and collaboration in an asynchronous way, web conferences that allow video, audio and written communication, and chat, where users can send messages and receive responses in real-time [41].

A Learning Management System is seen as a software that operates and encompasses many services that are meant to aid teachers in managing their lectures and courses [42], and they were created in order to monitor and evaluate students, give grades, to monitor course attendance or additional administrative actions that can be demanded by educational institutions [43]. These systems can be divided into two categories: open source-Moodle platforms, and commercial or proprietary, where platforms like Blackboard are included [44].

Designed to offer students, teachers, and administrators a system that can help them create an enhanced and customized learning climate, Moodle is considered a web-based flexible learning environment that facilitates collaboration between users [45]. Through these platforms, teachers can upload and supply students with information and resources to which they would not have had access during face-to-face classes, and students can easily share information, state their difficulties and receive feedback [46]. Thus, Moodle includes diverse features such as forums, chats, private messaging, and higher education institutions can use it as an additional method to traditional education, or for exclusively online learning [47].

Therefore, Moodle platforms are easy to access and use and they are known to have a positive impact on students' learning performance, Martín-Blas and Serrano-Fernández showing in their study that students who used Moodle during the academic year had better results and higher grades than students who weren't asked to use it [46].

## 2.3. Effectiveness, Benefits and Downsides of E-Learning

Through its complex characteristics and diverse features, E-learning can enhance the educational process. However, in order to positively influence collaboration and performance, teachers and students must know how to effectively integrate it into the teaching and learning process. The effectiveness of E-learning is determined, according to Tham and Werner, by three elements: institution—which refers to teachers knowing how to use the tools in order to enhance learning, how to interact with students and create a comfortable learning environment and how to creatively bring students closer and capture their attention, students—that may feel isolated because of the absence of physical colleagues, a case in which teachers should know how to establish connections and relationships with them, and technology [48].

Regarding online learning, comparative studies of face to face and online learning provided results that support its effectiveness in the educational field. Studies showed that when E-learning was applied, students were able to assimilate information as well as, or even better than students studying in the traditional way [49] and that online learning proved to be effective especially in the case of shy, easily intimidated, and slow learning students who usually do not have the courage to speak up and express themselves in the classroom [50].

Different from face-to-face learning, E-learning gained popularity mainly because of its flexibility in delivering education and accessing content and resources [51]. Thus, E-learning has great importance in the E-learning process for it has the ability to improve its quality, offering the possibility to personalize and adapt courses to the needs of the learners [22]. Due to its flexibility, E-learning eliminates barriers of space and time, the user can have access to a wide range of information, it facilitates collaboration, allows students to learn in their own rhythm, it motivates them to interact with their peers, discuss and exchange points of view and ideas [52]. Other studies mention as benefits the fact that online learning is faster, it saves time and money because it does not involve traveling [53], and the uploaded content is consistent and can be easily updated [54]. Furthermore, while studying the perception of students and teachers about E-learning, Al-Dosari found that from benefits such as accessibility, focus on the

students, flexibility, and collaboration, participants considered accessibility to be the most significant benefit of online learning [55].

Undoubtedly E-learning has many benefits, but some downsides of it can also be identified. Online learners may easily get distracted, lose focus, or miss deadlines, E-learning is dependent on technology: the internet and computers, which students may not have access to, and interruptions or other system errors may appear during courses [54]. For students, the ability to organize how they study and the amount of time spent on learning can sometimes result in decreased motivation, and the lack of physical interaction and presence of colleagues may determine students to feel isolated [56]. Drawbacks of E-learning can be seen in terms of physical health too. Because they spend many hours seated and in front of a screen, online learners and teachers may develop sight or back problems, and their outdoor activity may be reduced [57].

### 2.4. Online Learning during the Coronavirus Pandemic

Due to the unprecedented situation generated by the Coronavirus pandemic, the impact of the pandemic on education, universities, teachers and students, became a subject of great interest for researchers. Investigating students' perception about online learning during the Coronavirus, Allo showed that students had a positive attitude towards E-learning, considering it helpful and useful in the time of the crisis created by the pandemic [58]. A study [7] involving 424 universities around the world revealed that institutions were affected by the pandemic in terms of research, conferences, international mobility and education delivery, most universities stating that they had to adopt online learning and had to face many challenges, the most important being access to technology and teachers' ability to deliver online courses.

Even though some universities had used E-learning as an additional method before the Coronavirus pandemic, most of them were not ready for a full online experience. Thus, in order to continue to properly deliver education, optimization of the E-learning process is necessary. This optimization should also take into account student-teacher interaction, and the language used in the communication between students and teachers should be clear [59], but it should also contain specific terms for their field of study [60].

Furthermore, Sun et al., in their study on students' experience during online courses showed that students believe teachers should know how to adapt their lectures to the online environment, not just simply transfer online the information that was usually taught in the traditional way, and that they should give an adequate number of projects and assignments [61].

Moreover, Huang et al., identified seven important aspects that stand at the basis of online education and that have an essential role in optimizing learning in special circumstances like the ones created by the Coronavirus pandemic. These aspects involve: managing and developing internet infrastructure in order to avoid interruptions, especially during video-conferences; using friendly tools, that help students assimilate and understand information; providing reliable, interactive and diverse electronic resources; using social networks to build online communities for students in order to reduce feelings of isolation; using various effective techniques such as debates, or learning based on discovery and experience; providing services that help students and teachers learn about the latest policies adopted by universities and the government, and encouraging collaboration between these institutions [62].

## 3. Materials and Methods

### 3.1. Objectives and Research Questions

Most of the studies previously mentioned highlight a number of aspects related to the experience of students and teachers in the context in which online learning and implicitly, E-learning platforms were used as complementary tools to the traditional learning process. However, there are only a few studies that mention the exclusive use of E-learning platforms, as it happened during the pandemic when

universities were forced to use it and implement it as a main tool in the educational process. This study aims to illustrate students' perceptions regarding exclusively online learning through the E-learning platforms, in one of the countries that had little experience in this process before the pandemic.

Previous to the pandemic, in the Romanian higher education system, the use of E-learning platforms was scarce: few teachers were using the platform, and they were mainly using its basic functions such as uploading course material. The pandemic surprised most of the teachers but also students, who were very unfamiliar with online learning platforms, by forcing them to move, in a very short time, from traditional learning to exclusively online learning.

Therefore, we conducted an online questionnaire in order to examine students' perceptions regarding the ability of universities to provide knowledge in the context of exclusively online learning and to examine their attitude towards exclusively online learning.

In this study, the following research questions were addressed:

1.	What is the perception of students about the way universities managed to provide knowledge in the context of exclusively online learning?
2.	What is the perception of students about the ability to learn and assimilate information in the context of exclusively online learning?
3.	What is the perception of students about the use of E-learning platform in the process of exclusively online learning?

### 3.2. Participants

The population for the study was selected in a non-probabilistic way and was comprised of 762 students from two of the most important universities in Romania: Transilvania University of Brasov and West University of Timisoara. The sample consists of 405 participants from Transilvania University of Brasov and 357 participants from West University of Timisoara. The majority of respondents are female, mostly up to 22 years old, from urban areas, from Bachelor Primary Education programs (Table 1). However, 172 (22.6%) of the students were male and 590 female (77.4%), 274 (36%) were from a rural area, 488 (64%) from an urban area, and 685 (89.9%) were in the Bachelor Education programs and the others, 77 (10.1%) were in the Master Education programs.

**Table 1.** Sociodemographic characteristic of respondents.

| Variables | Category | Count | Percentage |
|---|---|---|---|
| Gender | Female | 590 | 77.4% |
| | Male | 172 | 22.6% |
| Residential environment | Rural | 274 | 36% |
| | Urban | 488 | 64% |
| Degree | Bachelor | 685 | 89.9% |
| | Master | 77 | 10.1% |
| Age | 18–22 years | 591 | 77.6% |
| | 23–25 years | 80 | 10.5% |
| | over 25 years | 44 | 5.8% |
| | n.r | 47 | 6.2% |
| University | Transilvania University of Brasov | 405 | 53.1% |
| | West University of Timisoara | 357 | 46.9% |

### 3.3. Data Collection Method

Data was collected online. The questionnaire was sent as a link on the Facebook pages of the two universities through the free application Google Forms, during the second semester of the 2019–2020 academic year. The research received the approval of the Ethics Commission in social research from Transilvania University of Brasov, Romania. The participants in the study received information at the beginning of the questionnaire about the purpose of the survey and the informed consent. Checking a

specific box, they approved the participation to the study. The e-mail addresses were not collected in order to respect anonymity and confidentiality. The average time needed to answer the questionnaire was 15 min.

### 3.4. The Research Instrument

For data collection, a non-standardized questionnaire was used. The questionnaire included items corresponding to the three research questions. For the first research question, the following items were included: frequency of technical problems in online learning (4 items related to the connection to the platform such as sound clarity, delayed message viewing, loss of signal during the audio/video conference), (Likert scale 5 = very frequently, 1 = did not use), frequency of using a set of 11 tools related to the E-learning platform (5-point Likert scale, where 1 = did not use, 5 = very frequently), compliance with the schedule (yes/no), balanced teaching style (more theory, less theory, balanced theory and practical tasks), assigning tasks to students compared to face-to-face learning (they have more free time, they have the same amount of tree time, they have less free time), obstacles encountered in the E-learning process (open question).

For the second research question the following items were used: opinion towards the use of the online environment for learning (5-point Likert scale, where 1 = to a very small extent, 5 = to a very great extent), preference for interaction with teachers during courses/seminars (microphone, chat), assimilation of information and online learning compared to face to face learning: perceived difficulty regarding the presentation of the seminar projects online (more difficult, the same dificulty, less difficult), proccessing information (more difficult, the same difficulty, less difficult), the type of course that facilitates information processing (audio, video, chat/forum), opinion towards online learning (5-point Likert scale, where 1 = not at all satisfied, 5 = very satisfied), preference for future learning in higher education (online, face to face, a combination between online and offline−hybrid).

For the third question of the research the following items were used: previous use of the E-learning platform (yes/no), frequency of use of 11 tools specific to the E-learning platform (5-point Likert scale, where 1 = did not use, 5 = very frequently), perception about the usefulness of the 11 tools in the process of learning (5-point Likert scale, where 1 = not at all useful, 5 = very useful), preference for the future use of various platforms.

The final part of the questionnaire contained a series of socio-demographic variables (gender, background, degree level: Bachelor/Master, field of study). This information was used only for the purpose of the descriptive analysis. The questionnaire and main items can be found in Table A1 (Appendix A).

### 3.5. Data Analysis

The data were analyzed using IBM SPSS Statistics (version 23). Answers to the open items were analyzed qualitatively, by recoding them into categories which describe the basic conditions for successful online learning (technical conditions, teachers' and students' technical skills, teaching in the online environment according to the teaching rules in the offline environment, teaching style, interaction with students/teachers). For the variables that were complexly measured, through several indicators (technical issues with the Platform, use of a large number of tools provided by the E-learning platform, usefulness of the tools provided by the E-learning platform) indexes [63] were made in order to synthetically measure the information from those indicators. For data analysis, descriptive statistics were made (percentages, mean, and standard deviation). Comparisons were made depending on the level of degree (Bachelor/Master) with the help of the Independent Samples *t*-Test and Chi square. In order to identify the correlation between students' perception about the use of the online environment in the teaching process and satisfaction towards learning in the online environment, the Spearman correlation coefficient was made. We considered a comparison between the levels of degree (Bachelor/Master) important because in Romania the profiles of the two categories of students are different. Master students are mainly students that are already employed, some of them with

families, and for them it is harder to be present during the courses even if the courses are held in the evening after their work shift has ended. We thought that, compared to Bachelor students, Master students could be fonder of E-learning because with E-learning it is easier for them to attend courses. In addition, because they have reached a more mature age, they have more capacity to assimilate information than Bachelor students.

## 4. Results

*4.1. What Is the Perception of Students about the Way Universities Managed to Provide Knowledge in the Context of Exclusively Online Learning?*

Universities, teachers and students were not prepared for the sudden shift to exclusively online learning and teaching, but they tried to find strategies to adapt and meet the new challenges (Table 2). Universities did not have the technical capacity to provide optimal conditions for online learning, 69.4% of the respondents complained that they frequently and very frequently encountered technical problems with the platforms provided by the universities (connecting to the platform, signal loss, delayed viewing of messages, the sound was not clear). For this reason, some teachers found alternative solutions by using other platforms, but this has generated stress among some students because there was no clear communication regarding when and where exactly the course will be held on other platforms (2.5% of students mentioned this aspect). Access to those platforms was sometimes poor, and there were connection problems here as well, especially when the number of students connected was high. Even more, students' lack of adequate technologies for participating in online learning has overlapped with these issues (poor internet connection, lack of laptops/computers, the mobile connection that partially offers access to resources provided by teachers and platforms), 14.8% of respondents mentioned these aspects.

Teachers did not have the necessary technical skills and they did not manage in such a short time to adapt their teaching style, or to properly interact with students in the online environment in order to assure high standards of the teaching process. The technical skills of teachers can be represented by their ability to use different functions offered by the E-learning platform in order to adapt their teaching style to the online environment, for example, using the video conference function where students can actively participate because teachers have the possibility to make them moderators. These technical skills also refer to the ability to present topics through screen sharing, to use synchronous chat during presentations, to offer students the possibility to work in teams during seminars, to post various links on the platform with reference to various sources of information, to make short videos for certain laboratories/seminars and to post them on the platform. Thus, some teachers managed to find solutions while others were not interested in making an effort to learn how to teach online. Thus, 86.4% of students stated that teachers frequently used a limited number of tools provided by the E-learning platform: they used only the basic tools which were almost mandatory for conducting the courses, and 30.6% of students declared they have used such instruments (Table 3. Moreover, to the open question, 15% of students mentioned that teachers did not have the necessary skills and they did not seem eager to improve their teaching skills in the online environment. Furthermore, 22.5% of students mentioned that the main issue they encountered was the lack of adaptation of the teaching style to the online environment, this having a negative impact on their ability to assimilate and understand the subjects taught during the courses. Regarding the courses, 32.8% of respondents declared that the schedule was not followed: teachers did not give breaks, classes did not start or end at the established hours.

In order for the teaching process to efficiently take place online, a balance between theory and practical tasks, as well as assigning tasks according to the amount of available time students have is necessary. Thus, 71.4% of students mentioned that the courses contained either too much theory or too many practical tasks, and 74.6% said they no longer have the same free time they had when teaching took place in the traditional way.

**Table 2.** Frequency distribution of indicators related to the university's capacity to provide knowledge in the process of exclusively online learning.

| Variables | Category | Percentage |
|---|---|---|
| Technical conditions | Technical problems while learning online (frequently and very frequently) [1] (open question) | 69.4% |
| | Use of multiple online platforms (depending on the teachers' preferences) (open question) | 2.5% |
| | Lack of student's adequate technologies (open question) | 14.8% |
| Teachers' technical skills | Diversified use of the tools offered by the E-learning platform (at least 9 out of the 11 instruments mentioned) [2] (closed question) | 46.1% |
| | Lack of teacher's technical skills (lack of interest for improving their skills, disorganization) (open question) | 15% |
| Adherence to the existing teaching rules in the offline teaching/learning system | Non-compliance with the schedule in the online environment (closed question) | 32.8% |
| Teaching style | Lack of adaptation of teaching style for the online environment (which generated difficulties of assimilation and understanding) (open question) | 22.5% |
| | Unbalanced teaching style (theory versus-practical tasks) (either just theory or just practical tasks) (open question) | 65.1% |
| | Unbalanced task allocation (students either have too little free time or too much) (closed question) | 70.2% |
| | Imbalance regarding the assignment of tasks that should be completed in a specified time (open question) | 8.8% |
| | Lack of clearly formulated requirements (closed question) | 4.4% |
| | Lack of ability to maintain students' attention (closed question) | 7.5% |
| Interaction with students/teachers | Lack of support from teachers in the learning process (deficient interaction) (closed question) | 10.2% |
| | Lack of interaction with peers/teachers (closed question) | 5.7% |

[1] A count index was built from 4 indicators that addressed various technical issues in conducting online courses (the category frequently and very frequently was counted). The maximum possible score is 4 and the minimum possible score is 0. Maximum score 4 = there were technical problems frequently and very frequently (connection difficulties, signal loss, delayed view of the messages, blurred sound), minimum score 0 = there were no such problems frequently and very frequently. [2] A count index was built from 11 indicators aimed at the use of various basic tools of the e-learning platform. (the category frequently and very frequently was counted). The maximum possible score is 11 and the minimum possible score is 0. Maximum score 11 = frequent and very frequent use of the 11 instruments mentioned. Minimum score 0 = use of none of the 11 instruments mentioned frequently and very frequently.

**Table 3.** Frequency distribution of indicators related to students' perception of the use of the E-learning platform.

| Variables | Category | Percentage |
|---|---|---|
| Previous experience | Previous use of the E-learning platform | 66.1% |
| The degree of use of the E-learning platform | Frequently and very frequently use of at least 7 of the 11 tools evaluated [1] | 86.4% |
| The usefulness of the E-learning platform | People who considered useful and very useful at least 7 out of the 11 instruments evaluated [2] | 68.9% |
| | Use of tools that involve collaborative learning | 30.6% |
| Intention to use | Preference for the E-learning platform | 28.6% |

[1] A count index was built from 11 indicators aiming at using the various basic tools of the e-learning platform (The category frequently and very frequently was counted). The maximum possible score is 11 and the minimum possible score is 0. Maximum score 11 = frequent and very frequent use of the 11 mentioned instruments. Minimum score 0 = use of none of the 11 tools mentioned frequently and very frequently. [2] A count index was built from 11 indicators aimed at the usefulness of the various basic tools of the e-learning platform. (The useful and very useful category was counted). The maximum possible score is 11 and the minimum possible score is 0. Maximum score 11 = usefulness (useful and very useful) of the 11 instruments mentioned. Minimum score 0 = usefulness of none of the 11 tools mentioned (useful and very useful).

Most students affirmed that they have less free time than they had before online learning due to the fact that teachers assigned them more tasks than usual. On the other extreme are the students that stated they have much more time than they used to because teachers do not request their attention or do not interact much with them. This is also confirmed by the answers to the open-ended question, where 8.8% of students said that the main problem was the lack of balance between the assigned tasks and the time teachers gave in order to solve them. In addition, some teachers did not mention clearly the requirements and expectations they had from students (4.4%) and failed to empathize with students or offer them support for their problems (10.2%). Some students also felt the lack of interaction with their teachers (5.7%) and their lack of focus and concentration in the online environment (7.5% of them mentioned this aspect).

*4.2. What Is the Perception of Students about the Ability to Learn and Assimilate Information in the Context of Exclusively Online Learning?*

Although the majority of students (66.1%) had used the E-learning platform previous to the pandemic (Table 3), their opinions regarding the use of the online environment for learning are divided. Some of them, (37.4%) consider it an appropriate environment for learning, some consider that is not very suitable (32%), and a third is undecided (30.6%). The same thing happens when it comes to the level of satisfaction towards their online learning experience: 39.1% being very satisfied and satisfied by the E-learning process, 31.9% were undecided, and 29% were dissatisfied (Table 4).

However, larger number of Bachelor students mention that online teaching is much more difficult than offline teaching ($\chi^2(2) = 21.44$, $p = 0.00$), (Table 5). Students that were more open to the use of the online environment for the teaching-learning process in general, had a higher degree of satisfaction with their exclusively online learning experience during the pandemic ($r_3 = 0.566$, df = 762, $p < 0.001$). The results also reveal that Master students were more open to use of the online environment in the learning process ($t(760) = 2.73$. $p < 0.001$), and they were more satisfied with their online experience ($t(760) = 4.18$. $p < 0.001$), (Table 6).

From the students' perspective, processing information is more difficult in the E-learning system (60.5%), while one third of them consider that even presenting seminar projects online is more difficult in the online environment (32.9%). However, if the courses were to be held in a videoconferencing system, this would increase the degree of understanding and information processing (73%). In this regard, it is interesting that during the courses/seminars, the vast majority of students prefer to interact more with teachers in writing, on chat/forum (52.4%), and prefer to use the microphone less.



**Table 4.** Frequency distribution of indicators related to student's ability to assimilate and learn while learning online.

| Variables | Category | Percentage |
|---|---|---|
| Interaction with teachers | Live answer in a video conference | 43.2% |
| | Offering an answer on chat/forum | 52.4% |
| | NR/NA (no answer) | 4.5% |
| Presentation of seminar projects/exercises online | It is harder | 32.9% |
| | It is easier | 33.3% |
| | It is the same | 33.7% |
| Processing information is easier when | The course is carried out audio | 19.9% |
| | The course is carried out audio and video | 73% |
| | The course is carried out on the chat/forum | 7.1% |
| Compared to offline teaching, online information processing is | Easier | 11.9% |
| | Harder | 60.5% |
| | The same | 27.6% |
| General opinion towards learning in the online environment | Dissatisfied | 29% |
| | Neither satisfied nor dissatisfied | 31.9% |
| | Very satisfied + satisfied | 39.1% |
| The online environment is appropriate for learning | Very little extent + little extent | 32% |
| | Neither little nor great measure | 30.6% |
| | Very great extent + great extent | 37.4% |
| Preference for online learning | Would choose the unfolding online courses | 10.6% |
| | Would choose the unfolding of face to face courses | 48.1% |
| | Would choose a combination between the online and offline courses | 41.3% |

**Table 5.** Chi-Square Tests related to degree.

| Variables | Pearson Chi-Square Value | df | Asymp. Sig. (2-sided) |
|---|---|---|---|
| Teaching online is more difficult | 21.44 | 2 | 0.00 |
| Preference for the E-learning platform | 4.13 | 1 | 0.04 |

**Table 6.** Independent Samples Test group of degree.

| Variables | | | | | *t*-Test for Equality of Means | | | | | | |
|---|---|---|---|---|---|---|---|---|---|---|---|
| | Group | N | Mean | S. D. | t | df | *p* | Mean Difference | Std. Error Difference | CI4 Lower | Upper |
| The online environment [1] | Bachelor | 685 | 3.04 | 1.19 | −2.73 | 760 | 0.00 | −0.39 | 0.14 | −0.67 | −0.11 |
| | Masters | 77 | 3.43 | 1.16 | | | | | | | |
| | Total | - | 762 | 3.08 | 1.19 | - | - | - | - | - | - | - |
| Online learning [2] | Bachelor | 685 | 3.04 | 1.18 | −4.18 | 760 | 0.00 | −0.59 | 0.14 | −0.87 | −0.31 |
| | Master | 77 | 3.64 | 1.12 | | | | | | | |
| | Total | - | 762 | 3.10 | 1.19 | - | - | - | - | - | - |
| Platform use [3] | Bachelor | 685 | 5.30 | 2.06 | 2.29 | 760 | 0.02 | 0.56 | 0.24 | 0.08 | 1.05 |
| | Master | 77 | 4.74 | 2.02 | | | | | | | |
| | Total | - | 762 | 5.25 | 2.06 | - | - | - | - | - | - |
| Platform usefulness [4] | Bachelor | 685 | 5.53 | 2.96 | −0.33 | 760 | 0.74 | −0.11 | 0.35 | −0.81 | 0.57 |
| | Master | 77 | 5.64 | 2.85 | | | | | | | |
| | Total | - | 762 | 5.54 | 2.94 | - | - | - | - | - | - |

[1] The online environment is appropriate for learning. [2] Satisfaction towards learning in the online environment. [3] Index, frequently and very frequently use of at least 7 of the 11 instruments evaluated. [4] Index, the self-perceived usefulness (useful and very useful) of at least 7 of the 11 instruments evaluated.

Therefore, taking into account the challenges they encountered, students consider that the most appropriate way to carry out the teaching-learning process is the traditional way, face to face, with them stating that the E-learning platform should be used as a complementary instrument to facilitate the educational process. Thus, 48.1% of students would prefer the teaching/learning process to take place exclusively face to face, 41.3% would choose a combination between online and offline courses and 10.6% would prefer exclusively online learning/teaching.

*4.3. Students' Perception of the Use of E-Learning Platform*

According to the Technology Acceptance Model (TAM), the actual use of a platform is influenced by the perceived ease of use of the instruments provided by the platform and by the perceived usefulness of those instruments. In this regard, it is notable that on the open question about difficulties encountered while using the platform, no student mentioned that the platform was difficult to use. Thus, 66.1% of students had previously used the platform and did not report any difficulty in using and operating the instruments it provides.

In the context of the Coronavirus pandemic, teachers were obliged to use only the E-learning platform provided by the university. The use of alternative platforms was allowed only to punctually solve the technical issues that arose due to the servers that were hosting the E-learning platform. The majority of teachers frequently used a large number of the tools offered by E-learning (over 7), (86.4% of students mentioned this aspect). However, a smaller percentage of students considered at least 7 of the 11 instruments useful (68.9%), (Table 3).

Even though the E-learning platform meets the basic conditions in order for students to turn their preference towards it, (ease of use, usefulness), students still prefer other platforms because of the small number of technical issues they had while using them. Unfortunately, students associated technical issues with the capacity of the platform itself, without making a difference between its use/usefulness and the capacity of the servers that were hosting it at that time. Thus, only 28.6% of respondents would choose the E-learning platform in order to carry out course-related activities, most of these students being enrolled in the Bachelor study programs ($\chi^2(1) = 4.13$, $p = 0.04$), (Table 6). Furthermore, it is also important to mention that the use of instruments offered by the platform was more reduced in the secondary education programs (Master programs) (t(760) = 2.29. $p = 0.02$), but perception about the usefulness of this platform is unitary regardless of the level of study programs (Bachelor or Master programs), (t(760) = −0.33. $p = 0.74$).

## 5. Discussion

In the context of the crisis created by the Coronavirus pandemic we assisted in two major changes in the Romanian higher education system: digitalization and the transition to a student centered E-learning process, changes that took place in a relatively short period of time.

With the digitalization, professors were forced to switch to a more student centered type of teaching, since E-learning platforms favor this type of teaching/learning [25]. Previous studies reveal that online learning platforms bring benefits for students when they are used as complementary tools for the traditional educational process [7]. Furthermore, most of the studies mentioned show that students generally have positive attitudes towards online learning [33–35], even though they sometimes encountered technical issues and consider that they process information better in the traditional courses [36]. In addition, other studies reveal multiple benefits of online learning such as better results in assimilating information [49,50], adapting courses to students' needs [22], flexibility [50], student centeredness [55], and removing the barriers of space and time, things that motivate students to join in on conversations and exchange opinions [52].

The quality of the educational process in the online environment depends on multiple factors, among which are: the level of training that teachers have in using technology, their teaching style, interaction with students, strategies used to capture students' attention, encouraging contact between students and faculty, collaborative learning, quick feedback, active learning, task time−encouraging students to allocate more time for completing tasks, high expectations–the teacher should communicate their expectations in order to encourage and motivate students, diversified learning, and technology application [26,27,48].

Taking into consideration these aspects, we analyzed the way in which teachers from two Romanian higher education institutions managed to provide knowledge during the Coronavirus pandemic. At the same time, we focused our paper on analyzing students' perception about their

experience during exclusively online learning, and what impact this type of learning had on their ability to learn and process information.

The present study shows that when learning exclusively online, some of the benefits and advantages previously mentioned diminish in value and disadvantages become more prominent. Students who responded to our questionnaire believe that exclusively online learning does not have beneficial effects on assimilating and processing information, that it is more difficult to study and be focused online, and that teaching is also harder. Furthermore, students also consider that presenting seminar projects is harder online, them not having the courage to speak up out of fear of being mocked or ridiculed. It is also very interesting that even though students used to actively participate in the offline seminar activities, online, very few have the courage to express their opinions or to write on the forum, because they feel more exposed.

In regard to the disadvantages of online learning, our study is in line with other studies [54,56,57]. Students easily get distracted and lose focus due to the fact that teachers do not have well implemented strategies in order to keep them focused, but also because of their lack of experience with this type of learning. In addition to these aspects, environmental disruptors such as the noise made by family members or neighbours and lack of adequate learning space also influence the amount of time for which students can concentrate while learning online. Another disadvantage that our study revealed is isolation. Students feel isolated because of the lack of interaction, especially with teachers, because they spend more time inside, in front of the computer, and because of the pandemic, which forced people to socially distance themselves from other people.

Our study is also in line with other studies [37,38] which suggest that students consider that the online educational process has less value than the traditional process, them preferring the use of E-learning platforms in combination with traditional, face-to-face teaching/learning. Furthermore, students believe the process of learning and assimilating information is poor in the online environment and this could result in poorer learning outcomes. Our study shows that it is possible for online learning to affect students' performance because respondents reported poor assimilation of information, especially when attending more difficult courses in which professors did not have well adapted methods of teaching. Furthermore, our research shows that universities were not ready to implement exclusively online teaching and learning, and this finding is consistent with the results of other studies [61], while other studies found that poor interaction with teachers was one of the main issues stated by students [9,11].

In the context of the crisis generated by the pandemic, the hierarchy of the reasons why students are reluctant to learn online is changing. Technical problems were the problems most frequently reported, them having a major role in decreasing students' motivation. Teachers' lack of technical skills is another important reason and this finding confirms the results of other studies conducted during the crisis [7]. The maladjustment of the teaching style to the online environment is the next reason, and on the last point, students mentioned poor commmunication and interaction with teachers. These last two reasons were generated on the one hand by the lack of technical skills, and on the other hand by the resilience to change and the lack of flexibility of some teachers to adapt in order to adequately provide knowledge in the online environment.

All these elements were reflected in students' perception about the quality of the online educational process, the overall score scale showing modest results: an average level of satisfaction.

Although there are some studies that indicate a positive attitude of students towards exclusively online learning during the crisis [58], the results of our research are consistent with recent studies conducted on students in Eastern Europe [64], which confirm a negative attitude of students towards online learning. However, the medium scores of the overall score scale regarding satisfaction towards exclusively online learning show that, in spite of all problems encountered, students, especially Master students, had the ability to relate these problems to the context of the pandemic when both teachers and students were forced to deal with a situation they hadn't faced before. Thus, some teachers did try

to learn, find solutions, offer support for students and adapt their teaching style to the new conditions, things that some students appreciated despite the existing technical difficulties.

Hence, considering the short period of time in which teachers had to adapt to the new teaching conditions, most of them managed to cope successfully with the challenges, but there is still room for improvement. In this regard, our findings reveal that the educational process was teacher-centered rather than student-centered, and when there had been an attempt to adopt a more student-centered process, students felt too much pressure due to the formidable number of tasks they were required to solve. A student-focused educational process involves assigning more responsibilities to students and more tasks, but unfortunately, because students were not accustomed to this type of learning they felt pressured, thus being more prone to develop negative attitudes towards online teaching and learning. Teachers used diverse tools while delivering courses online in order to make the course more attractive, but sometimes feedback from students was delayed, tasks were not concise, and teachers often failed to express clearly their expectations. The reason why the online educational process encountered so many issues is represented by the fact that the traditional way in which teachers used to deliver the practical part of the course was no longer suitable for the online environment. Thus, because they did not manage to rapidly adapt and come up with solutions, teachers created confusion and uncertainty among students.

Another important aspect that must be discussed is the interaction between students and teachers. According to our findings, students were affected by the lack of interaction with both teachers and peers. The E-learning platforms were not able to support video-conferences, except occassionally, in small groups and at special hours when the server was not overcrowded. However, for students, video-conferences were important because they could substitute the physical interaction with their teachers and colleagues. Some teachers managed to replace face to face interaction by assigning team projects, and some of them even used collaborative teaching tools.

When it comes to students' attitude towards the use of the E-learning platform, generally students consider the platform a useful tool for online teaching and learning. Still, due to the technical issues generated not necessarily by the platform, but by the servers of the universities that were hosting it, students would rather use other platforms. Students prefer platforms that allow multiple users to video communicate for longer periods of time, which do not generate so many technical problems, thus facilitating interaction between them and their teachers.

According to the Technology Acceptance Model (TAM) [28], the intention to use E-learning platforms is influenced by the perceived ease of use of the tools provided by the platforms and by the perceived usefulness of those tools. In this context, our findings reveal that students did not encounter difficulties while using the tools offered by the E-learning platform, them being intuitive and easy to manage. Even though the assessed instruments gained a high overall score scale regarding their usefulness in the learning process, only 28% of students would prefer to use these platforms, due to the technical issues it generated (issues such as signal loss, unclear sound, delayed viewing of the messages, that appeared because of the servers that were hosting it, because of the large number of students that were connected). In this regard, it is recommended for universities to develop strategies in order to solve these problems, because if such issues continue to persist, students may no longer be motivated to participate in the educational process.

Therefore, the results of our research revealed that just the perceived ease of use and perceived usefulness are not enough in order to determine students to use E-learning platforms, as other studies suggest [28–30]. Our results suggest that the Technology Acceptance Model (TAM) could be improved by taking into account some external factors such as technical conditions offered by the universities, students' technical conditions, teachers' teaching style, teachers' technical skills, and the interaction between students and teachers through the platform. If the TAM model can explain the intention to use E-learning platforms in the context in which the platform is used as a complementary tool for the traditional educational process, an improved version of the model could explain the intention to use it in the context of exclusively online learning.

## 6. Limitations

The study provides, from the students' point of view, relevant information regarding the way the educational process took place in Romanian higher education institutions in the context of the pandemic, information according to which the online teaching-learning process can be improved. However, the study also has some limitations. One limitation is represented by the fact that the sample was non-probabilistic and the research was conducted only on two Romanian universities. Thus, the results can not be generalized to the entire Romanian higher education system. Furthermore, the two universities had some experience with the E-learning platform prior to the Coronavirus crisis, even though only the basic tools of the platforms were used before the pandemic. It would be useful to broaden the sample to other universities from Romania, in order to be able to generalize results but also to make comparisons according to universities, fields of study, previous experience of universities with online learning, according to the existence of training programs for teachers during this transition period. Furthermore, it would be useful to conduct a longitudinal study that would allow us to see how universities adapted to teaching and learning exclusively online, if and how teachers adapted (teaching style, interaction with students), and if students' attitude towards online learning improved.

## 7. Conclusions

The results and findings of our study lead to two categories of implications: practical and theoretical implications. On a practical level, a series of useful recommendations for teachers can be outlined in order for them to succeed in increasing the quality of the educational process in the online environment.

The study offers a perspective regarding the way the educational process took place in a period of sudden and multiple changes in the Romanian higher education system. Thus, it is possible that after a longer period of adaptation and familiarization of students and teachers with the online environment, the quality of the educational process will improve, and that students' perception regarding online learning to be more positive and in line with other studies that we previously mentioned in this paper.

However, in order for the Romanian education system to properly and successfully adapt to online teaching and learning, a series of actions that can stimulate and facilitate its adaptation to this new type of teaching must be taken. In this regard, universities could develop training sessions for teachers or could develop programs whose role would be to stimulate teachers' performance and implicitly the quality of the educational process.

Technical issues are still the issues most difficult to solve, due to the capacity of the servers owned by universities. Surely, universities have made efforts in order to solve thesee problems and improve the way the E-learning platforms work. Still, students' technical problems remain poor internet connections, signal loss, lack of adequate digital devices, especially for students living in rural areas or students from families with low incomes. Universities could create programs to meet these types of needs and thus facilitate the learning process for students who find themselves in these situations.

In the Romanian higher education system, as well as in other systems [61], there is a need for concrete actions in order to improve and optimize the process of online teaching and learning, such as: improving teachers' technical skills, developing training programs meant to help teachers remodel and adapt their teaching style and the way they interact with students, to the online environment. The literature [64] provides some suggestions on how to adapt the educational process to the online environment, but the adjustments should be made according to the macro context at the level of each country, according to the profile and study field of the students.

From our point of view, the main challenges that the higher education system in Romania has to face today are: teachers' resilience to change and changing students' perception towards online learning. Training programs for teachers need to be developed in order to: help them adapt to the changes, to help them understand that the future of education in higher education systems involves the online environment, that it is unlikely that the system will return to how it was before the pandemic, and that online teaching is a new way to interact with students.

Teachers who are open minded, flexible and interested in developing themselves became self-taught and tried to improve their teaching skills. However, a certain segment of teachers still manifests resilience towards learning how to use new tools and they use, during the courses, only the basic functions of the E-learning platform. It is also important to mention that in the higher education system, it is more difficult for teachers to acknowledge that they do not know how to use certain tools provided by the platforms, which is why they do not ask for support in this regard. Still, only technical skills are not enough, teachers also have to adapt their methods of teaching to the online environment.

Regarding processing information while online learning, the results of the research show that for better assimilation of information it is necessary to use the videoconference function during courses, to balance the number of theory and practical tasks, and to assign students a proper number of practical tasks in order for them to avoid spending too many hours in front of the computer. It would be advisable for the tasks to involve teamwork to compensate for the lack of interaction in the online environment. Thus, this implies creativity in thinking and designing tasks that stimulate collaborative learning, but it also involves the need for technical skills in order to create and implement programs meant to improve interaction between students. Furthermore, it is necessary for teachers to quickly provide feedback on the tasks solved by students, to offer them support, to diversify the tasks offered to them, to use various teaching tools, to provide information in multiple ways (audio, video, text), and to find strategies to spark their interest and keep them focused during courses.

Another challenge would be to succeed in changing the way teachers interact and communicate with students. Students' lack of active learning, critical thinking skills, their lack of ability to debate and express their opinion, actions that the educational system did not foster or develop, are now becoming prominent in the higher education system in the process of online learning. If during face-to-face courses teachers still managed to find methods to encourage students to develop such skills, in the online environment it seems very difficult for teachers to find new methods. Students have strategies through which they avoid activities that require critical thinking, debate or that simply require them to state their opinion, and teachers have little control over them (the microphone is not working, the connection is bad, I cannot talk because I am also at work, or because there are other people in the room). We are not stating that such situations are not real, we simply highlight the fact that usually, students use them as excuses to avoid active participation in courses.

It is of utmost importance that teachers are available and open to students' needs in order to increase their engagement and involvement in the educational process (which is lower in the online environment). When students encounter technical difficulties, teachers should come up with solutions so that students could have access to the information provided during courses (recording and posting the course on the platform, offering supplementary materials), and if solutions can not be found, the least teachers could do is to be understanding and to not sanction students due to this type of issues. Answers like this: "there is nothing I can help you" or "it is your problem" only lower their motivation and increase students' frustration and the risk of dropping out.

Another implication of our findings can be found on a theoretical level. Starting from the conclusions of the studies conducted previous to the pandemic, the Technology Acceptance Model regarding the intention to use E-learning platforms could be developed and improved. The model could also include a series of external factors and could even be tested in the context of exclusively online teaching and learning.

**Author Contributions:** Conceptualization and project administration, C.C.; methodology, C.C., L.M.-S.; software, L.M.-S., investigation, L.G.Ț., C.S., M.C.B.; resources, L.G.Ț., C.S.; writing—original draft preparation, L.M.-S., M.C.B.; writing—review and editing, C.C., L.G.Ț., C.S. All authors have read and agreed to the published version of the manuscript.

**Funding:** This research received no external funding.

**Acknowledgments:** The authors would like to thank their colleagues, colleagues who disseminated the questionnaire to their students, and the students, who took the time to fill it in online.

**Conflicts of Interest:** The authors declare no conflict of interest.

## Appendix A

**Table A1.** Questionnaire.

| Research Questions. | Items |
|---|---|
| | U1_a. Which were the difficulties that you encountered while the courses/seminar were developed online) (very frequently, frequently, nor frequently, nor rarely, rarely, not at all.)<br><br>• Difficulties while connecting to the platform<br>• Losing signal during videoconferences<br>• Delayed visualization of messages communicated on the platform<br>• The sound is not clear (there are interruptions)<br><br>U1_b. What other difficulties did you encounter while the courses/seminars were developed online? (open question) |
| 1. What is the perception of students about the way universities managed to provide knowledge in the context of exclusively online learning? | U2. To what extent were used the following means available for teaching online courses/seminars on the E-learning platform? (not at all, rarely, nor rarely, nor frequently, frequently, very frequently)<br><br>• Audio conference<br>• Videoconference<br>• Documents posted on the platform (Word, Pdf, PowerPoint)<br>• Forum discussions<br>• Chat discussions<br>• URL addresses (to other web sources)<br>• Glossary of terms<br>• Course audio-video registered sequences<br>• Tasks in word/pdf format (that only the teacher could see)<br>• Task posted in databases (that were seen by the entire class)<br>• Documents were the whole class could work on in the same time |
| | U3. Taking into account the course schedule, do you consider that during online courses/seminars the start time of the course/seminar or the breaks were respected?<br><br>• Yes<br>• No<br>• I don't know/I'm not answering |

**Table A1.** *Cont.*

| Research Questions. | Items |
| --- | --- |
| 1. What is the perception of students about the way universities managed to provide knowledge in the context of exclusively online learning? | U4. Regarding the structure of the courses taught online, their content included:<br><br>• More theory than practical tasks<br>• More practical tasks than theory<br>• The same amount of theory and practical tasks<br><br>U5. Ever since courses / seminars began to be held online, you have for individual study and project preparation:<br><br>• Less time<br>• More time<br>• Nor less time, nor more time |
| 2. What is the perception of students about the ability to learn and assimilate information in the context of exclusively online learning? | L1. To what extent do you consider the online environment to be suitable for education and training at university level?<br><br>1. In a very small extent<br>2. In a small extent<br>3. Nor in small, nor in great extent<br>4. In great extent<br>5. In very great extent<br><br>L2. From the perspective of student-teacher interaction, when you are in a position to provide an answer to the teacher's questions, you prefer to:<br><br>• Answer live during a videoconference<br>• Offer a written answer on forum/chat<br>• I don't know/I'm not answering<br><br>L3. Compared to the presentation of projects in classrooms, when you present a project online<br><br>• You find it harder to present<br>• You find it easier to present<br>• You find it nor easier, nor harder to present |

**Table A1.** *Cont.*

| Research Questions. | Items |
|---|---|
| | L4. Taking into account the acquisition and assimilation of the information transmitted by the teacher, in the online environment it is easier for you to process the information when:<br><br>• The course is held audio<br>• The course is held with video and audio<br>• The course develops in writing, on forums/chats |
| | L5. Compared to face-to-face taught courses/seminars, during online courses/ seminars for you it is:<br><br>• Easier to assimilate information<br>• Harder to assimilate information<br>• Nor easier, nor harder to assimilate information |
| 2. What is the perception of students about the ability to learn and assimilate information in the context of exclusively online learning? | L6. Overall, how satisfied are you with your experience with the online teaching system on the E-learning platform?<br><br>1. Not at all satisfied<br>2. Not really satisfied<br>3. Nor satisfied, nor dissatisfied<br>4. Fairly satisfied<br>5. Very satisfied |
| | L7. If you had the possibility to choose, you would rather prefer:<br><br>• For the courses/seminars to be held online<br>• For the courses/seminars to be held face to face<br>• A combination between online and offline courses/seminars<br>• I don't know/I'm not answering |

**Table A1.** *Cont.*

| Research Questions. | Items |
|---|---|
| | E1. Have you used the E-learning platform before suspending offline courses?<br><br>• Yes<br>• No |
| | E2. How useful were these means of teaching courses / seminars on the E-learning platform? (not at all useful, somewhat useful, nor useful nor useless, useful, very useful, it's not the case/it wasn't used) (usefulness of the E-learning platform- TAM)<br><br>• Audio conference<br>• Videoconference<br>• Documents posted on the platform (Word, Pdf, PowerPoint)<br>• Forum discussions<br>• Chat discussions<br>• URL addresses (to other web sources)<br>• Glossary of terms<br>• Course audio-video registered sequences<br>• Tasks in word/pdf format (that only the teacher could see)<br>• Task posted in databases (that were seen by the entire class)<br>• Documents were the whole class could work on in the same time |
| 3. What is the perception of students about the use of E-learning platform in the process of exclusively online learning? | E3. On which of these platforms would you have preferred the online courses/seminars to be held? (intention to use the E-learning platform-TAM)<br><br>• Skype<br>• E-learning<br>• Facebook<br>• Zoom<br>• Avaya<br>• Google Meet |
| | U1_b. What other difficulties did you encounter during online courses / seminars? (open question) (how easy it is to use the E-learning platform: we focused our attention on answers regarding the difficulties of using E-learning, which are actually related to its tools and not the capacity of the servers or the internet connection) |

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
