# Peer review of "Online Teaching and Learning in Higher Education during the Coronavirus Pandemic: Students’ Perspective"

_sustainability, doi:10.3390/su122410367_

Round 1

Reviewer 1 Report

The study addresses an important issue in dealing with online teaching and learning during the COVID-19 pandemic. Without a doubt, it was challenging as it was sudden and unforeseen for every educational institution. I´m sure,  In a normal situation, the current study would not be relevant and even conducted but it's now more than ever important as it is now due to the COVID-19 pandemic giving a new perspective to the field. 

My comments:

1) The body text needs re-organization. The paper has sections that consist of only one or two sentences (for example see lines 98 to 99; 220 to 223; or 322 to 330).

2) It is not clear how the research instrument was constructed. Mainly how it relates to the TAM model that is discussed with results. Also, I suggest adding a table that summarises the instrument and how and which items support research questions or are derived from the TAM model. The latter is already described within the text but it is not easy to read and synthesize from the reader's point of view.

3) Please explain why it was important to compare BA and MA students. Did you have a theoretical expectation for it?  

4) I recommend keeping consistency in terminology (online learning/ online teaching or e-learning; COVID-19 or coronavirus.  Although one might think that e.g., online learning or e-learning is well-known but still there are potential readers who might not know. What do mean by technical skills? Add some examples.5) Typos  (e.g., lines 350 or 391).

I really enjoyed reading the article as it has a certain historical retro perspective. Thank you!

Author Response

Response to reviewer 1 comments

We kindly thank the reviewer for taking the time to review our article. We found the suggestions very helpful for improving our article and thus we tried to comply with all the changes suggested by the reviewer. We made the changes according to each of the four comments made by the reviewer using the “Track Changes” function integrated in Microsoft Word, having active the option “All Markup.” Thus, the numbers of the lines that we will further provide in order for our changes to be identified easier in the revised manuscript, can be viewed precisely by selecting the option “All Markup” from the  “Track Changes” function in the Word document. Moreover, in order to view correctly all the numbers of the lines where we made changes, the whole text should be read without accepting the changes one at a time.

Before responding to every point we want to mention that, since the reviewer ticked the box “English language and style are fine/minor spell check required”, we spell checked again our text and changes are visible throughout the manuscript because we made them with the Track Changes options.

Point 1: The body text needs re-organization. The paper has sections that consist of only one or two sentences (for example see lines 98 to 99; 220 to 223; or 322 to 330)

Response 1: Thank you for pointing this out. The suggestion is very helpful and we made changes in this regard. However, due to other changes that we had to make to the text, the line numbers in the revised manuscript differ from the initial manuscript. Thus, lines 98 to 99 are now 128 – 129, lines 220 to 223 are now 254 to 257, and lines 322 to 330 are now 359 to 367.

Lines 128 to 129 contained only one sentence: ”However, in the technology era, one of the main challenges of universities is the integration of innovative E-learning systems so as to reinforce and support both teaching and learning”, and in order to correct this error we put the sentence in continuation to the paragraph before it, in continuation to the lines: 124 – 127.

Lines 254 to 257, contained only the following sentences: ’”Through its complex characteristics and diverse features, E-learning can enhance the educational process. However, in order to positively influence collaboration and performance, teachers and students must know how to effectively integrate it into the teaching and learning process.” We reorganized the text and linked these sentences to the next paragraph: ”The effectiveness of E-learning is determined...” With ”Track Changes”, and the option ”All markup active”, the new formed paragraph starts at line 254 and ends at line 262.

We linked lines 359 to 367, which contained three paragraphs with only one or two sentences. Since they all referred to Data collection method, linking them did not change the meaning of the new formed paragraph that can be found at lines: 359 to 366.

Point 2: It is not clear how the research instrument was constructed. Mainly how it relates to the TAM model that is discussed with results. Also, I suggest adding a table that summarizes the instrument and how and which items support research questions or are derived from the TAM model. The latter is already described within the text but it is not easy to read and synthesize from the reader's point of view.

Response 2: We thank you for the suggestion. We took the suggestion very seriously and in order to provide a clearer view on the instrument that we used in order to conduct the research we created a table that summarizes the instrument. We inserted the table in Appendix A, which starts at line 806. In the table, questions related to each of our research questions are presented and among them are questions that associate the questionnaire with the Technology Acceptance Model (TAM). The model shows that intention to use E-learning platforms is influenced by the attitude people have towards using the platform, attitude that is influenced by perceived usefulness and ease of use. Thus, our questionnaire also contains questions that refer to the intention to use the platform and its usefulness, such as: “E2. How useful were these means of teaching courses / seminars on the E-learning platform? (not at all useful, somewhat useful, nor useful nor useless, useful, very useful, it’s not the case/it wasn’t used) (usefulness of the E-learning platform- TAM)”, or “E3. On which of these platforms would you have preferred the online courses/seminars to be held? (intention to use the E-learning platform-TAM).”

Point 3: Please explain why it was important to compare BA and MA students. Did you have a theoretical expectation for it?

Response 3: The comment is very helpful and we thank you for suggesting us to provide an explanation for the comparison we made between Bachelor and Master Students. In order to comply with this suggestion we inserted at lines 415 – 422 (the line numbers with the All markup option active, not the simple markup), the following explanation: “We considered important a comparison between the levels of degree (Bachelor/ Master) because in Romania the profiles of the two categories of students are different. Master students are mainly students that are already employed, some of them with families, and for them is harder to be present during the courses even if the courses are held in the evening after their work shift has ended. We thought that, compared to Bachelor students, Master students could be fonder of E-learning because with E-learning it is easier for them to attend courses. In addition, because they have reached a more mature age and they have other capacity to assimilate information than Bachelor students.”  

Point 4: I recommend keeping consistency in terminology (online learning/ online teaching or e-learning; COVID-19 or coronavirus.  Although one might think that e.g., online learning or e-learning is well-known but still there are potential readers who might not know. What do mean by technical skills? Add some examples.5) Typos  (e.g., lines 350 or 391).

Response 4: The recommendation is very helpful and we are grateful for pointing out that consistency in terminology must be kept. We kept the E-learning formulation. In this regard, we replaced in the text online teaching/online learning with E-learning. While viewing the text with the All markup option active, the lines where we made the change described above are: 87,108,110,11,118,221,257,271,303,383,494,503.

Next, we kept the term Coronavirus throughout the text because we found it more suitable since the term also appears in the title of the article. Thus, at lines 16 and 41 in the revised manuscript, we changed the term COVID 19 with the term Coronavirus.

Further, to address the suggestion about explaining what do we mean by technical skills and adding some examples, within the paragraph that used to start at line 391, and that now, in the revised manuscript starts at line 451, we added beginning with line 453 the following text: “The technical skills of teachers can be represented by their ability to use different functions offered by the E-learning platform in order to adapt their teaching style to the online environment. For example, using the video conference function where students can actively participate because teachers have the possibility to make them moderators. These technical skills also refer to the ability to present topics through screen sharing, to use synchronous chat during presentations, to offer students the possibility to work in teams during seminars, to post various links on the platform with reference to various sources of information, to make short videos for certain laboratory/ seminars and to post them on the platform.”

Reviewer: I really enjoyed reading the article as it has a certain historical retro perspective. Thank you!

Response from authors: We want to thank again the reviewer for making time to review our article and for providing such useful suggestions.  

Reviewer 2 Report

I would like to congratulate the authors for a well-written, structure and sound article. It will be of great interest for the educational community duing these times of pandemics.

Author Response

Response to reviewer 2 comments

Reviewer: I would like to congratulate the authors for a well-written, structure and sound article. It will be of great interest for the educational community during these times of pandemics.

Response from authors: We are very thankful for the kind and encouraging words that the reviewer addressed us. We also want to thank the reviewer for taking the time to read and review our article. 

Reviewer 3 Report

It is indeed necessary scientific article.

Some items to pay attention to:

  1. Line 56, 146, 159, 224-229 - dash (-) joins the word.
  2. Line 141 - empty space.
  3. From my point of view Introduction should be more clearly presented. Lots of things are mentioned and no clear understanding of main problems.
  4. Introduction and Literature Review contain information that is not new and it is recommended to cut it.
  5. The graphics and tables are absent - that makes difficult to analyze data (complicates the perception of the presented material).

Author Response

Response to reviewer 3 comments

We are very grateful for the time the reviewer spent on reviewing our article and for providing suggestions in order to improve it. We sincerely appreciate the pertinent suggestions that the reviewer made, and we tried to address the suggestions in the best way possible.

Before providing a response to every point made by the reviewer we want to mention that we made the changes while having active the “ Track Changes” function that Microsoft Word provides, and the option “All Markup”. Thus, changes we made and the line numbers that we will provide in our responses can be best viewed by activating the “All Markup” option.

Point 1: Line 56, 146, 159, 224-229 - dash (-) joins the word.

Response 1: The comment is very helpful and we thank the reviewer for pointing out lines that contain errors. In order to comply with the suggestion, at every line mentioned, we added a space between the dash (-) and the word that joined it. However, due to other changes made to the text, in the revised manuscript the number of the initial lines, mentioned by the reviewer, changed. In this regard, line 56 became line 80, line 146 became line 176, line 159 became 189, and lines 224 -229 became 257-262. ( with the “All Markup” option active).

At line 80, the incorrect phrase was “health- providing”, and we added a space before the dash: “health - providing”.

At line 176, the incorrect phrase was “task time- encouraging”, and we added a space before the dash: “task time - encouraging.”

At line 189, the incorrect phrase was “system- which”, and we added a space before the dash: “system - which”.

At lines 257-262, the incorrect phrase from the line 258 was “institution- which”, and we added a space before the dash “institution - which”. The incorrect phrase from the line 260 was “students – that”, and we added a space before the dash “students - that”.

Point 2: Line 141 - empty space.

Response 2: As described in the response 1 that we provided, other changes that we were required to make to the manuscript changed the numbers of the lines. Thus, in the revised manuscript and with the “All markup” option active, line 141 is now line 171. Taking into account the comment of the reviewer, we removed the empty space that was between the end of the sentence and the beginning of the next one.

Point 3: From my point of view Introduction should be more clearly presented. Lots of things are mentioned and no clear understanding of main problems.

Response 3: We kindly thank the reviewer for this comment. We believe this suggestion is very important for improving the article and we tried our best in making the introduction more clearly. In order to comply with the suggestion, several steps were made. Firstly we rearranged the sources comprised in the Introduction in order for the Introduction to be more clear and cursive. Next, we reformulated some phrases in order to link the rearranged source and highlight the main issues that we wanted to address in the Introduction.

 In the revised manuscript, the numbers of the sources that we rearranged changed. The sources were not changed, only their number. Considering the source numbers from the initial manuscript, they were rearranged in the Introduction in the following order: [1], [3], [2], [4]. [9], [13], [5], [7], [12], [8], [9], (other information from source 9), [11].

In the revised manuscript the numbers of the sources changed in the following way:

[1] remained [1] - it has the same place in the text as it had in the initial manuscript; [2] was interchanged with [3]; [4] remained [4] - it has the same place in the text as it had in the initial manuscript; [9] became [5], [13] became [6], [5] became [7]; [7] became [8]; [12] became [9]; [8] became [10]; [11] remained [11] - it has the same place in the text as it had in the initial manuscript, but we added more information from the study.

Now, the Introduction firstly mentions that the pandemic brought changes to the teaching and learning process in higher education and that governments took measures in order for higher education institutions to continue to deliver courses. The second paragraph highlights the fact that the pandemic made E-learning a necessity and that during the pandemic students may have different perceptions about E-learning than they had previous to the pandemic.

The third paragraph focuses on presenting the benefits and advantages of E-learning while the fourth and fifth paragraphs focus on presenting the obstacles and challenges that both universities and students had to face during the pandemic. Paragraphs six and seven describe which ideas, also presented in the literature, stood at the basis of the research, then they present the aim of the paper, the elements that we focused our research on, and why this type of research can contribute to the development of the E-learning process.

Thus, besides rearranging sources, we also added information in order to link better the sources and highlight the main issues. We made the following changes:

Lines 44 – 47, we added the text “This paradigm shift could generate changes in students’ perception of this way of teaching and their perception might be different from the one found in studies previous to the pandemic. Thus, through this paper, we tried to capture the existence of such changes.”, to highlight that the pandemic might influence students’ perception about E-learning.

Lines 48 – 51, the text of source [4] was reformulated. In the revised manuscript, the text emphasize better the benefits of E-learning.

Lines 58 – 60,  we added “This might contribute to better communication with the students and in spite of some inherent challenges brought by this time of crisis, E-learning might enhance the learning process for students.”, as a remark that strengthens the idea that E-learning has benefits.

Lines 61 – 62, we reformulated and added “when using E-learning platforms there are also some elements that might be considered obstacles in students’ process of learning”, in order to emphasize that in the next paragraphs the obstacles and challenges of E-learning will be discussed.

Lines 65 – 71. Between sources [8] and [9] we added “Nonetheless, these obstacles can be overcome with the help of teachers who should adapt their teaching strategies to the needs of students. In order to do so, experience and knowledge about teaching in the online environment are necessary. Thus, we believe that these challenges and disadvantages could be more prominent while the educational process takes place exclusively online. This might happen due to the lack of teachers’ experience in using E-learning and due to the short time in which they had to adapt their teaching style to the new conditions.”, to express our view on the challenges and obstacles faced by students during the pandemic.

Line 71- 73 present information from source [9] and after we added “Thus, it can be inferred that students and teachers were not ready for an entirely online experience.”, to highlight our opinion about the fact that students and teachers were not prepared for exclusive E-learning. (one of the main issues)

Lines 86-90: we added more information from the study presented by source [11], to describe challenges that students had to face. “On the other hand, students also had to face challenges and a study focusing on students’ perspective on E-learning identified that among the main challenges students encountered were accessibility, connectivity, lack of appropriate devices, social issues represented by the lack of communication and interaction with teachers and peers”.

Lines 101-107 were reformulated to present better the elements that stood as basis of our research.

Moreover, as a consequence of the rearrangement of the sources, information from lines 90 – 102 was removed and formulated according to the new arrangement we presented at the beginning of this response.

The order of the sources in Introduction can be seen in the revised manuscript in the References section, ( lines 810 – 854 sources) as well as the changes made in order to rearrange them.

Point 4: Introduction and Literature Review contain information that is not new and it is recommended to cut it.

Response 4: We are grateful for the suggestion and we tried to comply with it by removing some of the sources that were old and that provided no new information. Before describing the sources we removed, we mention that because of the rearrangement that was made in order to make the introduction more clear, (as suggested at Point 3), but also because of the removal of some sources, the numbers of all the sources in the revised manuscript changed. Thus, we removed the sources that in the initial document had the numbers [6], [10], [29] and [42].

Source [6] Pollard, E.; Hillage, J. Exploring E-Learning. IES Report 376.The institute for employment studies. 2001,ISBN: 1-85184-305-1. Changes made in order to remove the source can be seen in the revised manuscript. (lines 833 – 834, page 22, with the ”All markup” option active).

Source [10] Hrastinski, S. Asynchronous and synchronous e-learning. Educause quarterly, 2008, 31, 51-55. Changes made in order to remove the source can be seen in the revised manuscript.( line 847, page 23, with the ”All markup” option active).

Source [29] Govindasamy, T. Successful Implementation of e-Learning Pedagogical Considerations. Internet High. Educ. 2001, 4, 287–299, https://doi.org/10.1016/S1096-7516(01)00071-9. Changes made in order to remove the source can be seen in the revised manuscript.( line 886 – 887, page 23, with the ”All markup” option active).

Source [42] Brown, B.W.; Liedholm, C.E. Can Web Courses Replace the Classroom in Principles of Microeconomics? Am. Econ. Rev. 2002, 92, 444–448. Changes made in order to remove the source can be seen in the revised manuscript.( line 919-920, page 24, with the ”All markup” option active).

 Source [33] Davis, F.D. User acceptance of information technology: system characteristics, user perceptions and behavioral impacts. Int. J. Man-Mach. Stud. 1993, 38, 475–487, https://doi.org/10.1006/imms.1993.1022, from the initial manuscript was old too, but the information it presented was essential to our research. Thus, we removed the source [33], but we we replaced it with a newer source co-authored by the author of [33], that presented the same information that we found in the previous source. In the revised manuscript, the new source (with the same information) has the number [30] Venkatesh, V.; Morris, M. G.; Davis, G. B.; Davis, F. D. User acceptance of information technology: Toward a unified view. MIS quarterly. 2003, 425-478 https://doi.org/10.2307/30036540. It can be found at line 895, page 24. (with the ”All markup” option active) The changes made in order to remove source [33] and replace it with source [30] can be seen in the References section.( lines  894-897, with the ”All markup” option active).

Point 5: The graphics and tables are absent - that makes difficult to analyze data (complicates the perception of the presented material).

Response 5: We thank you for pointing out the missing tables and graphics. The suggestion is very useful in order to improve our article. Before describing how we addressed the comment made by the reviewer we want to mention that when we initially submitted the manuscript, we submitted the tables we created as Supplementary Materials. However, in order to make it easier for the reader to analyze the data and in order for the text to be easier to understand, we now inserted the tables within the text. The 6 tables that present the results of the research were integrated into the Results section, and another table that summarizes the research instrument was integrated into Appendix A section.

Table 1 presents “Sociodemographic characteristic of respondents”, it can be found at line 356 (page 8), and we inserted it right after the text related to it.

Table 2 presents the “Frequency distribution of indicators related to the university’s capacity to provide knowledge in the process of exclusively online learning”, it can be found at line 440 (page 10), right after the text related to it.

Table 3 presents “Frequency distribution of indicators related to student’s ability to assimilate and learn while learning online”, and it can be found at line 516, (page 12). The table contains data related to the text above the table. (lines 489- 495).

Table 4 presents the “Chi-Square Tests related to degree”, and can be found at line 520, (page 13).

Table 5 presents the “Independent Samples Test group of degree”, and can be found at line 547, (page 14).

Table 6 summarizes the questionnaire and presents items (questions that students were required to answer) according to our research questions and the Technology Acceptance Model that we also considered while constructing the questionnaire. The table can be found in Appendix A, line 806, page 19. (with the ”All markup” option active)

We thank again the reviewer for spending time on providing us suggestions to improve the article.

Round 2

Reviewer 3 Report

It is a paramount research nowadays.